# Enhancement of Antioxidant and Hydrophobic Properties of Alginate via Aromatic Derivatization: Preparation, Characterization, and Evaluation

**DOI:** 10.3390/polym13152575

**Published:** 2021-08-02

**Authors:** Smaher M. Elbayomi, Haili Wang, Tamer M. Tamer, Yezi You

**Affiliations:** 1Hefei National Laboratory for Physical Sciences at the Microscale, CAS Key Laboratory of Soft Matter Chemistry, Department of Polymer Science and Engineering, University of Science and Technology of China, Hefei 230026, China; smaher@mail.ustc.edu.cn (S.M.E.); hfwanghl@mail.ustc.edu.cn (H.W.); yzyou@ustc.edu.cn (Y.Y.); 2Polymer Materials Research Department, Advanced Technologies, and New Materials Research Institute (ATNMRI), City of Scientific Research and Technological Applications (SRTA-City), New Borg El-Arab City, Alexandria P.O. Box 21934, Egypt

**Keywords:** alginic acid, antioxidant, anti-inflammatory, ABTS, DPPH

## Abstract

The preparation of bioactive polymeric molecules requires the attention of scientists as it has a potential function in biomedical applications. In the current study, functional substitution of alginate with a benzoyl group was prepared via coupling its hydroxyl group with benzoyl chloride. Fourier transform infrared spectroscopy indicated the characteristic peaks of aromatic C=C in alginate derivative at 1431 cm^−1^. HNMR analysis demonstrated the aromatic protons at 7.5 ppm assigned to benzoyl groups attached to alginate hydroxyl groups. Wetting analysis showed a decrease in hydrophilicity in the new alginate derivative. Differential scanning calorimetry and thermal gravimetric analysis showed that the designed aromatic alginate derivative demonstrated higher thermo-stability than alginates. The aromatic alginate derivative displayed high anti-inflammatory properties compared to alginate. Finally, the in vitro antioxidant evaluation of the aromatic alginate derivative showed a significant increase in free radical scavenging activity compared to neat alginate against DPPH (2,2-diphenyll-picrylhydrazyl) and ABTS free radicals. The obtained results proposed that the new alginate derivative could be employed for gene and drug delivery applications.

## 1. Introduction

Oxidative stress has attracted significant scientific attention as an intermediary in the etiology of many human diseases. The consequence of an imbalance between the production of oxidants and endogenous antioxidants to counteract is oxidative stress because of deficiency of antioxidants or growing reactive oxygen species (ROS), including superoxide anions, hydroxyl radicals (HO^·^), hydrogen peroxide (H_2_O_2_), singlet oxygen (^1^O_2_), reactive nitrogen species (RNA), and reactive sulfur species (RSS) development [1,2]. Free radicals are assumed to play a significant role in the disease’s development and the ageing process, for example, cell signaling, apoptosis, ion transportation, and gene expression because of free radical unpaired electrons, which have shown high activity in reacting with other molecules with a view of neutralization [3]. Antioxidants are the first line of protection (defense) against the detrimental consequences of free radical damage, and the maintenance of optimal health is essential by various mechanisms of action. Antioxidants are oxidation inhibitors, even at low concentrations; subsequently, antioxidants play a crucial role in the body. Generally, antioxidants possess their activity in two basic ways, either by inhibiting the formation of ROS/RNS or neutralizing/scavenging ROS/ RNS by reacting and demolishing the reactive radicals to convert them to a harmless substance [4,5]. Nowadays, researchers are trying to use naturally based antioxidants, such as polysaccharides.

Natural polymers are more influential than synthetic polymers because they are renewable, economic, nontoxic, easily accessible, stable, biodegradable, biocompatible, and prone to chemical modification compared to expensive synthetic ones that have shown toxicity and environmental issues [6,7,8,9]. Polysaccharides are the most abundant natural biopolymers, which have received increased attraction as influential materials in different biomedical fields, potentially due to their special intrinsic features [10,11,12,13,14]. Polysaccharides have different functional groups and show differential physicochemical properties and fundamental biological activities, making them attractive materials in many pharmaceutical areas, like tissue engineering and drug delivery [15,16,17,18]. Subsequently, the utilization of these natural carbohydrates in particular pharmaceutical applications is increasing [19,20,21]. 

Alginate, a polysaccharide, has emerged as one of the most extensively investigated biopolymers due to its unique characteristics and versatility [22,23]. Alginates have consisted of two hexuronic acids: β-D-mannuronic acid (M) and α-L-guluronic acid (G), connected by 1–4 bonds [24,25]. These units are distributed randomly in a linear chain. They can also be coordinated as homogeneous MM or GG blocks, as well as heterogeneous or alternating MG blocks [25,26]. Alginates have also been studied in terms of their biological properties. Researchers have demonstrated that alginate polymer has antioxidant properties, which increases by breaking the polymer chain [27,28]. These compounds have substantial free radical scavenging and antioxidant activity, like phenolic content (the Folin–Ciocalteu reagent) [29], the activity of antioxidants assessed through DPPH radical scavenging activity (1,1-diphenyl-2- picrylhydrazyl) [29,30,31], and antimicrobial activity as well [31]. Borazjani et al. [30] demonstrated that alginates can be obtained the stable form of DPPH in a dose-dependent manner. Besides, in a redox linked colorimetric reaction, alginates were also capable of reducing materials from ferric to ferrous. Additionally, alginates have anti-inflammatory properties that rely on containing D-mannuronic acid; specifically, the D-mannuronic acid named M2000. It is a recently developed nonsteroidal anti-inflammatory drug (NSAID) with major inflammatory disease-controlling functions [32]. In the NSAID drugs group, the D-mannuronic acid molecule is recognized as a molecule with less toxicity and the lowest molecular mass [33]. Numerous studies have demonstrated the significance of the D-mannuronic acid molecule’s therapeutic and anti-inflammatory possibility in various animal model experiments [34].

Alginates are utilized frequently because of their rheological properties [35,36], biocompatibility, biodegradability, and low toxicity [37,38]. Alginates also have unique biological and pharmacological characteristics [38]. They mainly perform as gelling agents, stabilizers, and thickeners in the food and pharmaceutical industry [24,39]. Moreover, these biopolymers are commonly described as healthy food elements owing to their anticancer and prebiotic properties [40]. 

Another use of alginates in the food industry is food packaging [41]. Alginates are also used in a broad range of biomedical engineering projects due to their favorable properties, including tissue engineering, controlled drug release, cells encapsulation, and immunostimulatory effects [25,41,42,43].

In this study, we report the new derivatization of alginate using benzoyl chloride. The prepared benzoyl alginate derivative was identified using different characterization tools. 

Additionally, the antioxidative and anti-inflammatory properties of the synthesized benzoyl alginate derivative were evaluated compared to this property of alginate.

## 2. Materials and Methods

### 2.1. Materials 

Alginic acid was purchased from (ACROS, ORGANICS), benzoyl chloride and 2,2-azinobis-[3-ethylbenzothiazoline-6-sulfonic acid] diammonium salt (ABTS; purum, ≥99%; meilunbio) were used as received from Sigma Aldrich. We used K_2_S_2_O_8_(p.a. purity, max 0.001% nitro-gen; Merck, Germany) and 2,2-diphenyl-picrylhydrazyl (DPPH, 97%, Kaihang pharmaceutical). Hydrochloric acid (Diachem Chemicals), ethanol (99.8%, Fisher Chemical), acetone (≥99.9%, Brand Chemical), albumin bovine fraction V (MP Biomrdicals, LLC), trypsin, fresh human red blood cells, and dimethyl sulfoxide were also utilized. Deionized high-purity grade H_2_O, with a conductivity of ≤0.055 S/cm, was produced using the TKA water purification system (Water Purification Systems GmbH, Niederelbert, Germany).

### 2.2. Methods

#### 2.2.1. Preparation of Phenolic Alginate Derivative 

The synthesis of aromatic alginate derivative (MAlg) was carried out by creating a reaction of hydroxyl alginate groups with benzoyl chloride, as illustrated in Figure 1. Briefly, 2.0 g of alginic acid was dissolved in 80.0 mL of DMSO. Then 3.2 g of benzoyl chloride was added at a weight ratio equal to 45% of the Alg monomer-mol% to the solution, which was previously taken in a 2-necked flask mounted on constant temperature water bath at 80 °C for 12 h. The flask was connected to a condensation column. Subsequently, the benzoyl alginate derivative was precipitated using 200.0 mL acetone, centrifuged, washed twice with acetone on a sintered glass filter, and dried under reduced pressure.

#### 2.2.2. Fourier Transfer Infrared Spectroscopy (FT-IR)

The reaction between alginic acid and benzoyl chloride was confirmed by identifying the functional groups in the chemical structure of alginic acid and its derivative (discs of 1–2 mg sample was added to 200 mg KBr) using a FT-IR spectrophotometer (Shimadzu FTIR-8400S, Kyoto, Japan). The data were construed utilizing the IR solution software over a range of wavelengths between 4000 and 400 cm^−1^, using 30 scans at a resolution of 4 cm^−1^.

#### 2.2.3. Elemental Analysis

The alginic acid and its aromatic derivative were investigated by elemental analysis using a 2400 CHN analyzer (PerkinElmer, Waltham, MA, USA). Measurements were taken in duplicate.

#### 2.2.4. Nuclear Magnetic Resonance Spectroscopy (NMR)

Samples were run as approximately 5% *w*/*v* solutions in DMSO in 5 mm tubes using tetramethylsilane (TMS) as the internal standard. The 1H-NMR spectra were evaluated at ambient temperatures (approximately 301 K) utilizing a Joel JNM-FX 100 Fourier transform NMR spectrometer operated at sweep widths of 250 ppm and 10 ppm, respectively. Peak intensities were identified through electronic integration.

#### 2.2.5. Thermogravimetric Analysis (TGA)

Thermal analysis of alginate and alginate derivatives samples (~6 mg) were performed using a thermogravimetric analyzer device (Shimadzu TGA −50/50H, Kyoto, Japan) within temperatures from ambient −800 °C at a heating rate of 10 °C/min under nitrogen flow (20 mL/min).

#### 2.2.6. Differential Scanning Calorimeter (DSC)

Differential scanning calorimetric analysis of alginate and alginate derivative samples (~5 mg in sealed Al-pan) was carried out using Differential Scanning Calorimeter device (Shimadzu DSC–60A, Japan) in a temperature range of ambient −500 °C at a heating rate of 10 °C/ min under nitrogen flow (10 mL/min). All thermal measurements were performed on single samples and used for comparative purposed against control samples. 

#### 2.2.7. Scanning Electron Microscopy (SEM) 

Scanning electron images of samples of alginate and its derivatives were recorded using a scanning electron microscope (Joel Jsm 6360LA, Kyoto, Japan). The evaluated samples were fixed on a specimen mount with carbon paste. The surface of the samples was coated with a thin layer of gold to eliminate the poor conductivity of the sample’s current before testing.

#### 2.2.8. Biodegradability 

The aim of this test was to assess the samples’ responses to phosphate buffer saline (PBS, pH = 7.4) and its influence and impact of hydrolysis on the samples. The samples’ weight loss was evaluated by weighing the sample regularly before and after biodegradation testing at every regular time interval (10 day). The samples’ weight loss with time was utilized to determine the biodegradation amount in the PBS environment. The percentage of the samples’ weight loss was estimated by deducing the identified weight at each interval compared to its original weight after washing with water. The samples were dried before and after at 50 °C overnight to ensure the accuracy of the results.

#### 2.2.9. Antioxidant Activity 

ABTS Radical Scavenging Assay 

ABTS radical scavenging assay was investigated using a modified version of the procedure from Re et al. [44]. ABTS radicals (ABTS) were produced through partial oxidation of ABTS with K_2_S_2_O_8_ by incorporating an aqueous solution of K_2_S_2_O_8_ (3.30 mg) in water (5 mL) with ABTS (17.2 mg). The resulting bluish–green radical cation solution was incubated overnight in the dark below 0 °C. Subsequently, a volume of 1 mL of the ABTS solution was diluted to a final volume of 60 mL with distilled water. Alginate or alginate derivative solution (10 mg/mL) was added to 2.0 mL of the ABTS solution and carefully mixed and maintained for 5 min at room temperature before assessing the absorbance at 730 nm. The percentage of ABTS scavenged was evaluated according to the following formula: $$\mathrm{Inhibition}\;\%=\frac{\left[\mathrm{AC}-\mathrm{As}\right]}{\mathrm{Ac}}\times 100$$
where Ac is the absorbance of ABTS solution without polymer solution and As is the absorbance of the sample with ABTS solution. Samples were measured in triplicate spectrophotometric measurements.

##### DPPH Radical Scavenging Activity

The alginate and alginate derivative solution antioxidant activity was assessed with minor modifications using the DPPH method. Forty milligrams of 2,2-diphenyll-picrylhydrazyl were dissolved into 100 mL of methanol, respectively. Therefore, DPPH solution was diluted with 100 mL of deionized water. The working solution of DPPH was produced through diluting 200 mL stock DPPH solution with 800 mL of water/methanol (50:50, *v*/*v*) mixture. The 0.5 mL of DPPH solution was combined with 1 mL of alginate and alginate derivative. The reaction mixture was then shaken well and incubated in the dark for 30 min at room temperature. The decolorization of DPPH dye was evaluated at 527 nm using a UV–VIS spectrophotometer. The percentage inhibition of radicals was calculated using the following formula: $$\mathrm{Inhibition}\;\%=\frac{\left[\mathrm{AC}-\mathrm{As}\right]}{\mathrm{Ac}}\times 100$$
where Ac is the absorbance of DPPH solution without extract and As is the absorbance of the sample with DPPH solution.

##### In Vitro Anti-Inflammation Activity

Analysis of the effect of phenolic alginate derivative on heat-induced bovine serum albumin (BSA) denaturation assay was performed with minor modifications using a method developed by Chandra et al. [45]. The reaction mixtures contained different concentrations (100, 200, 300, 500 μg/mL) of synthetic phenolic alginate and native alginate, 1% *w*/*v* BSA and phosphate buffer saline (PBS, pH 6.4) separately, whereas PBS was utilized as a control. The reaction mixtures were maintained in the incubator at 37 °C for 20 min and the temperature was raised to maintain the samples at 70 °C for 5 min. After cooling, the inhibition was assessed at 660 nm using UV-visible spectrophotometer (Model Ultrospec 2000). The control described 100% protein denaturation. The percentage of inhibition of BSA denaturation was calculated using the following equation:
%Inhibition of BSA denaturation = 100 × (1 − A2/A1)

where A1 = absorbance of the control and A2 = absorbance of the test sample.

#### 2.2.10. Hemolysis Test

Fresh human red blood cells (RBC) were introduced in tubes containing heparin sodium, centrifuged to isolate and extract the RBCs from the plasma, then washed three times with 150 μL PBS solution to gain the concentrated RBC. Concentrated RBC (50 μL) in 10 mL PBS was resuspended to get RBC stock solution. The alginate or alginate derivative, H_2_O, and RBC solution were mixed at ranging concentrations from 10 to 100 μg/mL in separate tubes and incubated at 37 °C for one hour. Then, the cells were collected by centrifugation for 5 min. The supernatant’s absorbance was analyzed at 541 nm using a microplate reader, utilizing 0% hemolysis and 100% hemolysis controls, respectively. The cell hemolysis percentage was calculated by using the following formula: Hemolysis (%) = (Abs-Abs0)/(Abs100-Abs0) × 100, where Abs, Abs100, and Abs0 represent the absorbance scales of the sample, the 100% hemolysis solution, and the 0% hemolysis solution, respectively. All hemolytic experiments were carried out in triplicates.

#### 2.2.11. Investigating the Cytotoxicity 

Human normal fetal lung cell line (Wi-38) was used to investigate toxicity of the alginate and alginate derivative according to method described by Mosmann (1983). Human Wi-38 was maintained in DMEM medium (Bend, OR, USA) containing 10% fetal bovine serum (GIBCO, Waltham, MA, USA). This cell line was sub-cultured for 2 weeks before assay using trypsin EDTA (Lonza, USA). Their viability and counting were detected by trypan blue stain and hemocytometer. Wi-38 cells were seeded in a 96-well culture plate as 1 × 104 cells per well and incubated at 37 °C in 5% CO_2_ incubator. After 24 h for cell attachment, cells were treated with serial amounts of alginate and alginate derivative. After 72 h incubation in 5% CO_2_ incubator, 20 µL of MTT solution (5 mg/mL) was added to each well and incubated at 37 °C for 4 h in a 5% CO_2_ incubator. MTT (MO 63118, Sigma-Aldrich, St. Louis, MO, USA) solution was removed and the insoluble blue formazan crystals trapped in cells were solubilized with 150 µl of 100% DMSO at 37 °C for 10 min. The absorbance of each well was measured with a microplate reader (BMG LabTech, Offenburg, Germany) at 570 nm to estimate cell viability. The doses (EC50 and EC100) that caused 50% and 100% cell viability of the tested compounds were estimated by the Graphpad Instat software. 

## 3. Results

### 3.1. FT-IR

To strengthen antioxidant and anti-inflammatory properties of alginate, an aromatic alginate derivative as a new derivative was prepared and characterized as demonstrated in Figure 1 hydroxyl groups of alginates were coupled with benzoyl chloride to develop the appropriate aromatic alginate derivative.

The FT-IR spectrum of alginate and aromatic alginate derivative (MAlg) is presented in Figure 2. In the 3600–1600 cm^−1^ region, three bands showed up: a broad band centered at 3374.5 cm^−1^ designated to hydrogen bonded O–H stretching vibrations and for the aromatic alginate derivative, the weak signal at 2928 cm^−1^ because of C-H stretching vibrations, and the asymmetric stretching of carboxylate O-C-O at 1735 cm^−1^. The band at 1415.8 cm^−1^ can be because of deformation vibration with contribution of O–C–O symmetric stretching vibration of carboxylate group [46,47]. The new band at 1431 cm^−1^ may be attributed to C=C stretching vibration band.

The weak bands at 1358.9, 1173.7, and 1093.7 cm^−1^ may be assigned to C–C–H and O–C–H deformation, C–O stretching, and C–O and C–C stretching vibrations of pyranose rings, respectively; the band at 1034.84 cm^−1^ can also be because of C–O stretching vibrations. The most frequently discussed region in carbohydrates is the fingerprint or anomeric region (931.6–738.7 cm^−1^) [46,48]. The spectrum demonstrates a band at 931.6 cm^−1^, which was appointed to the C-O stretching vibration of uronic acid residues, and one at 878.6 cm^−1^, assigned to the C-H deformation vibration of mannuronic acid residues. The band at 804.3 cm^−1^ appears to be typical of mannuronic acid residues [49,50].

The aromatic derivative of alginate was also identified by elemental analysis, as presented in Table 1. The decrease of the C/H ratio in the aromatic derivative from 7.385% to 6.67% indicated the substitution process of the aromatic benzoyl group to the polysaccharide backbone.

### 3.2. Thermal Gravimetric Analysis

Figure 3 shows the TGA curves of alginic acid (Alg) and its aromatic derivative (MAlg). It could be seen that three consecutive weight loss steps were observed. The first stage of weight loss for both Alg and MAlg involved loss of about 16.05% for Alg and 8.2% for MAlg from their initial weight, which was probably due to elevation of piping and intermolecular water molecules from the polymer. Decreases in the intermolecular water in alginic acid by modification were due to the new aromatic group’s hydrophobic nature on alginic acid. 

The second weight loss, with a rapid decomposition between 230 and 350 °C, was due to the complex disintegration processes, including saccharide rings and macromolecule chains of alginic acid and its derivatives. The shift of decomposition temperature from 289 °C in alginic acid to 345 °C in the aromatic derivative indicate a protectivity role of aromatic groups against thermal decomposition. 

### 3.3. Differential Scanning Calorimetry

The thermal characterization was completed by DSC experiments, the results of which are depicted in Figure 4. The results obtained are in good agreement with the findings of TGA. Some changes in the number of exothermic/endothermic peaks were observed between different membrane constituents and their positions. The thermogram gave a broad endothermic peak centered at about 100 °C for Alg and 75 °C for MAlg, resulting from the release of moisture content in the membrane. The increase of the endothermic peak of Alg over MAlg can be explained by the ability of Alg to hold more moisture molecules than MAlg. The exothermic peaks around 235 °C and 240 °C may have been due to the decomposition of alginic acid’s intermolecular chains. MAlg displayed the same behavior in a combination of many exothermic peaks that can be attributed to the decomposition of part of the Alg that formed an aromatic bond with the benzene ring.

### 3.4. Morphological Analysis

SEM micrographs for Alg and MAlg are illustrated in Figure 5. From the shown micrographs, it can be seen that the roughness of tested surfaces increased with the functionalization of alginic acid with an aromatic ring. This phenomenon can be attributed to heterogeneous molecules between the polymeric alginic acid chains and immobilization of the phenolic group with glucose hydroxyl groups on the repeating polysaccharide, which distorts the internal order of chains and influences the polymer crystal structure.

### 3.5. Nuclear Magnetic Resonance Spectroscopy (NMR)

Figure 6 demonstrates the HNMR of alginic acid and its modified aromatic derivative. The left figure indicates the pure alginic acid with its characteristic signals. The signal at δ 2.52 ppm was attributed to the proton in C4, where signals at δ 3.43 ppm and δ 3.73 ppm refer to a proton in C3 and C2. The signal at δ4.5 ppm was attributed to the proton attached to C4. On the other hand, the proton in C1 was shifted to 4.88 ppm. Hydroxyl protons attached to C3 and C4 were observed at δ 3.6 ppm.

Functionalization of alginic acid with benzoyl chloride generated a new peak at 7.4–7.5 for aromatic protons.

### 3.6. Biodegradability

The degradation of alginic acid and aromatic alginate derivatives were more fully evaluated under phosphate buffer saline (PBS) at pH 7.4 conditions. Figure 7 demonstrates the weight loss percentage of alginic acid and aromatic alginate derivative in PBS medium at a constant 37 °C for several days. Generally, the weight loss percentage of alginic acid and its aromatic alginate derivatives enhanced as a degradation time function. The weight loss percentage of the aromatic alginate derivative slightly increased and reached 33% after 9 degradation days. This observation demonstrates that the degradation was a hydrolytic degradation of the aromatic alginate derivative. Nevertheless, a substantial weight loss of almost 50% of aromatic alginate derivatives was identified at day one of degradation in PBS medium. Almost 100% of the molecular weight reduction happened during 9 days. The degradation rate of the aromatic alginate derivative increased by introducing a new function group to alginic acid. This may be ascribed to the abundant carboxylic groups in alginic acid. Esterified alginates are more susceptible to β-elimination in comparison with other alginate derivatives because of the increasing of the electron-attracting influence of the carbonyl group in C6 by esterification, which increases the rate of removal of H-5 in the first step of the β- elimination degradation mechanism [51].

### 3.7. Antioxidant Experimental 

Depending upon the chemical reactions involved, the antioxidant capacity assays can be classified into two categories: [52] hydrogen atom transfer (HAT) reaction-based assays and [53] single electron transfer (ET) reaction-based assays. HAT- and ET-based assays are designed to evaluate the radical (or oxidant) scavenging capacity rather than a sample’s preventive antioxidant capacity. HAT-based assays contain oxygen radical absorbance capacity (ORAC) [52,53] and total radical trapping antioxidant parameters (TRAP) [54]. HAT-based methods typically consist of a synthetic free radical generator, an oxidizable molecular probe, and an antioxidant. ET-based assays evaluate an antioxidant’s ability to reduce an oxidant, which changes color when reduced. The degree of color change is associated with the antioxidant concentrations of the sample. ET-based assays contain the total phenol assay by Folin–Ciocalteu reagent (FCR), Trolox equivalence antioxidant capacity (TEAC) [44], ferric ion reducing antioxidant power (FRAP), “total antioxidant potential” assay using a Cu (II) complex as an oxidant, and DPPH [55].

Various antioxidant mechanisms have been generally described for carbohydrates. Throughout their theoretical research of the radical scavenging ability of carbohydrates, Hernandez-Marin et al. [56] indicated that HAT fundamentally occurs from carbon-bonded hydrogens and that ET is less likely to happen. A possible antioxidant mechanism of carbohydrates is typically not considered, as they do not frequently comprise aromatic rings or double bonds. In their research of several antioxidants, Peshev et al. [57] mentioned that compounds containing a carbon–carbon double bond were preferable antioxidants. The double bond offers radical addition, which becomes the favorite radical reaction over ET and HAT.

#### 3.7.1. Antioxidant Capacity Assay (ABTS Assay)

The ABTS assay was utilized to quantify saccharide’s ability to scavenge free radicals. Figure 8a demonstrates the decolorization of ABTS^•+^ by appending alginic acid and the aromatic alginate derivative. The color of ABTS^•+^ was significantly decreased by 19% and 93% under the influence of alginic acid and aromatic alginate derivatives, respectively. Increasing ABTS^•+^ decolorization from alginic acid to the aromatic alginate derivative may be associated with enhancing the ability of the alginate derivative to scavenge ABTS^•+^. Presence of the benzoyl nucleus along with the polymer backbone facilitates its tendency to donate the electron. ABTS^•+^ possesses a bluish–green color with maximum absorbance values at 734 nm, which rapidly disappear through electrons from antioxidants [58,59,60]. Figure 8a shows a spectacular concentration-dependent improvement in the decolorization effect of the tested polymer. Higher free radical scavenging potentials were observed at higher concentrations The increase in antioxidant activity by concentration could be associated with forming more functional groups, such as hydroxyl, carbonyl, and carboxyl groups in alginic acid (and benzoyl group in esterified derivative) [61]. The electron-donating activity of the aromatic alginate derivative was substantially higher than that of alginic acid itself. Additionally, ABTS^•+^ scavenge activity shows results in terms of time. The color of the ABTS^•+^ of alginic acid and the aromatic alginate derivative was decreased by 15–70% with time (30 min) (Figure 8b). The aromatic alginate derivative was observed to scavenge ABTS^•+^ in a time and concentration manner.

#### 3.7.2. Free Radical Scavenging Activity of DPPH (2,2-Diphenyll-Picrylhydrazyl)

The scavenging of hydrogen radicals is an essential antioxidant mechanism. DPPH possesses a hydrogen free radical and demonstrates a characteristic absorption at 517 nm [62]. After encountering the proton radical scavengers, the DPPH purple color solution fades quickly [63]. Therefore, DPPH was utilized in this study to evaluate the proton scavenging activity of the alginate derivative.

Free radical scavenging activities of DPPH of alginic acid and the aromatic alginate derivative are introduced in Figure 8c. Lower DPPH values described the alginate compared to the aromatic alginate derivatives. Alginate and aromatic alginate derivative in higher concentrations had more outstanding scavenging capabilities. Scavenging abilities of alginic acid and aromatic alginate derivative against hydroxyl radical were deduced from hydrogen atom abstraction reaction between the hydroxyl groups in the polysaccharide unit and the hydroxyl radical.

The aromatic alginate derivative exhibited the highest radical scavenging activity, followed by the alginate; in general, the DPPH radical scavenging efficiency increased with the concentration of aromatic alginate derivative. For instance, as a concentration of phenolic alginate derivative increased from 0.1 to 1.0 mg/mL, scavenging activity toward DPPH radicals raised from 75.6% to 96.8%, indicating that aromatic alginate derivative showed stronger scavenging activity toward DPPH than alginic acid. 

### 3.8. In Vitro Anti-Inflammation Assays 

The denaturation of protein facilitates autoantigenic production, leading to inflammation of rheumatic diseases [64]. The inhibition mechanism of protein denaturation utilizing anti-inflammatory materials at high temperatures has been identified [65,66]. Data in Figure 9a exhibited that 100–500 μg/mL alginic acid and aromatic alginate derivatives inhibited heat-induced BSA denaturation in a concentration-dependent manner. The aromatic alginate derivative significantly showed higher inhibition of heat-induced BSA denaturation than alginic acid. A rise in the inhibition percentage was accomplished when the concentration of the alginate derivative was increased. It is obvious that aromatic alginate derivative effectively inhibited the BSA protein denaturation, even at lower concentrations (100 μg/mL). Nevertheless, a higher inhibition of BSA (55%) was documented at 500 μg/mL. Protein denaturation is an inflammatory consequence. 

### 3.9. Invitro Blood Compatibility

Blood compatibility is a significant biological characteristic for new biomedical material because of direct contact with blood. The hemolysis test is considered a reliable and straightforward measure for hemocompatibility estimation [67], and a hemolysis percentage lower than 5% is appropriate [68]. Some studies have demonstrated that increasing molecular weight and a material positive charge density can increase its hemolytic property [69,70]. Furthermore, data in Figure 9b showed that the 10–100 μg/mL alginic acid and aromatic alginate derivative stabilized erythrocyte membrane against hypotonicity-induced hemolysis in a concentration-dependent manner. The aromatic alginate derivative was not significantly toxic for RBCs, even at high concentrations. Moreover, it was mentioned that alginic acid does not cause any toxic effects, corroborating literature reports [71]. The hemolysis process can influence not only the positive charge density but also spatial structure and molecular weight. 

### 3.10. Investigating the Cytotoxicity

Cellular toxicity is an integral approach to assess the cytocompatibility of various polymeric materials [72]. Consequently, an MTT test was utilized to estimate the in vitro toxicity of alginic acid and aromatic alginate derivatives towards normal fetal lung cells. A series of alginate and aromatic alginate derivatives with concentrations ranging from 0.01563–0.5 mg/mL were used in the cells. From the data in Table 2, it is clear that the cellular toxicities correlated with a rise in the values of the tested polymer concentration. All concentrations of the tested material exhibited no significant variances compared to the control cells, reporting cell viability ratios of 66.7 ± 0.026% and 62.868 ± 0.026% in the presence of 0.5 mg/mL of alginic acid aromatic alginate derivative, respectively.

Furthermore, the morphology of the cells exhibited no notable changes after culturing with the tested polymer solution for two days. Conversely, more than 95% of cells were destroyed after two days of treatment with cisplatin.

Based on high EC50 and EC100 values that denoted high safety of the tested compounds on normal cell viability, alginate and aromatic alginate derivatives were the safest compounds compared to other tested compounds (Table 3 and Figure 10). Overall, the cytotoxicity data implied that the tested polymers had good biocompatibility, supporting their implementation as a potential drug and gene delivery.

## 4. Conclusions

Aromatic alginate derivative was prepared and characterized. Expected results revealed significant changes in its features. Therefore, it can be described as following:

The FT-IR analysis and HNMR confirmed the chemical structure of aromatic derivatization.

The thermal stability of aromatic alginate derivatives was strengthened with the introduction of the benzoyl group. 

SEM analysis revealed that the prepared aromatic alginate derivative exhibited increased surface area due to roughness and porosity of the surface compared to alginate.

The anti-inflammatory evaluation showed that aromatic alginate derivatives demonstrated better anti-inflammatory activity than alginate.

Antioxidant assessment demonstrated that the aromatic alginate derivative had a lower propensity to scavenge hydrogen free radicals. Furthermore, it is an electron donor, which was exhibited by ABTS and DPPH assay.

## Figures and Tables

**Figure 1 polymers-13-02575-f001:**
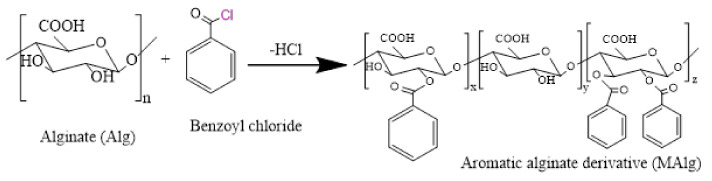
Schematic preparation of aromatic alginate derivative.

**Figure 2 polymers-13-02575-f002:**
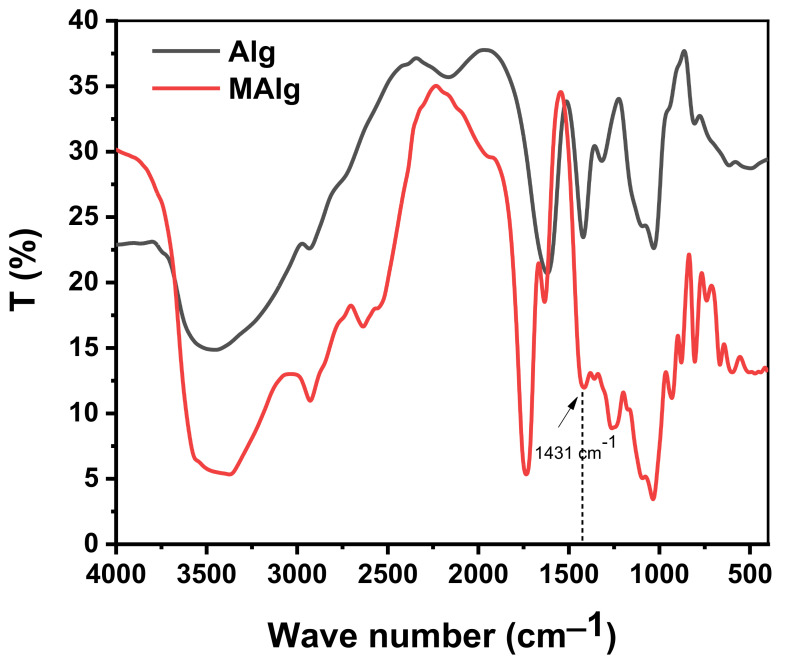
FT-IR of alginic acid and its aromatic derivative.

**Figure 3 polymers-13-02575-f003:**
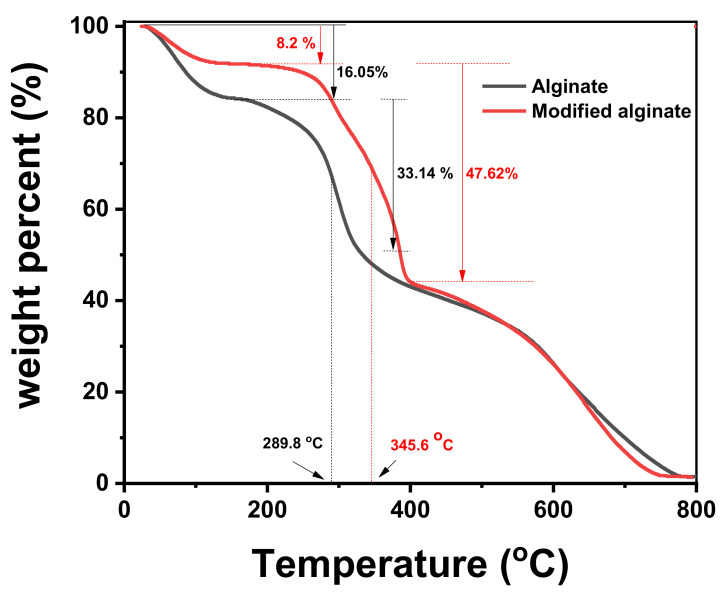
TGA of alginic acid and its aromatic derivative.

**Figure 4 polymers-13-02575-f004:**
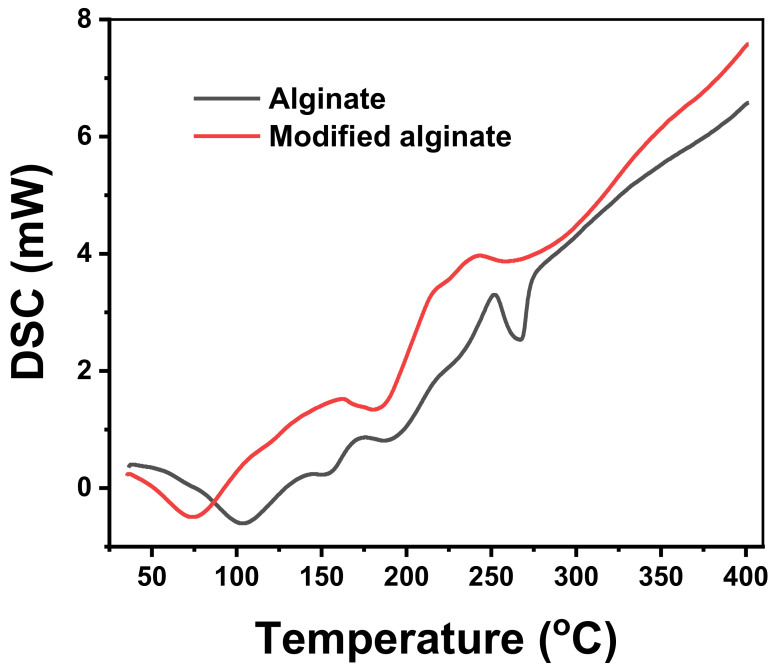
DSC of alginic acid and its aromatic derivative.

**Figure 5 polymers-13-02575-f005:**
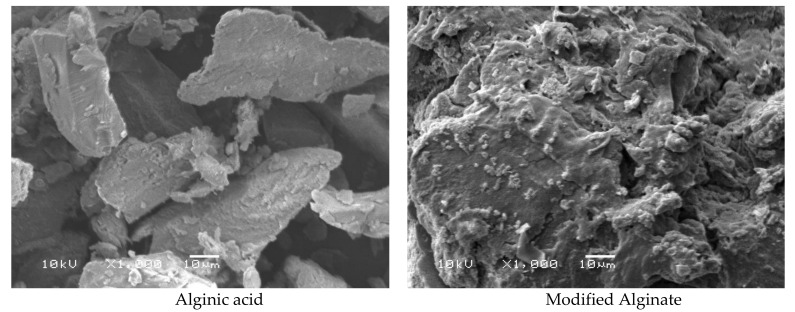
SEM images of alginic acid and its aromatic derivative.

**Figure 6 polymers-13-02575-f006:**
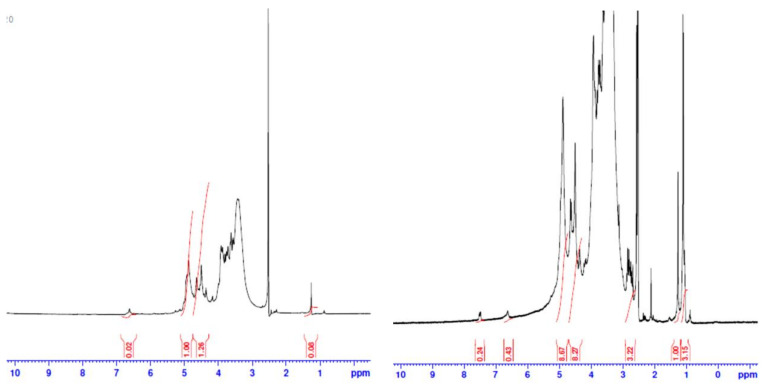
H NMR spectrum of alginic acid (in **left** side) and its aromatic derivative (in **right** side).

**Figure 7 polymers-13-02575-f007:**
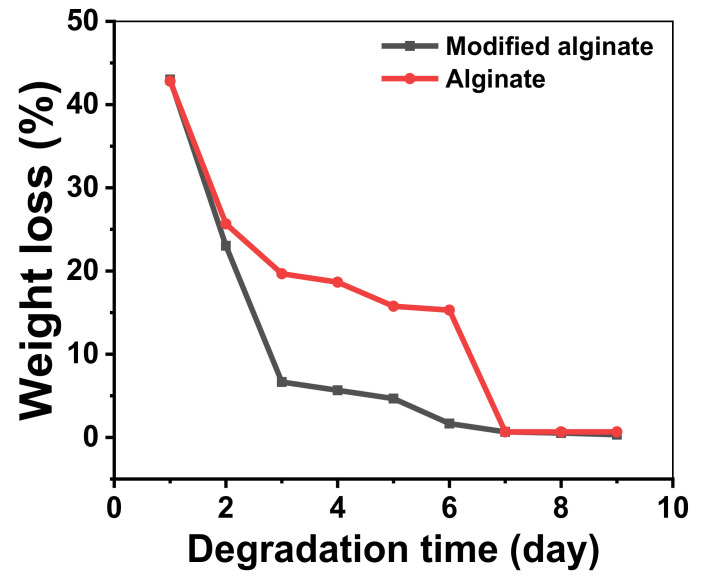
Hydro degradability of alginic acid and its aromatic derivative.

**Figure 8 polymers-13-02575-f008:**
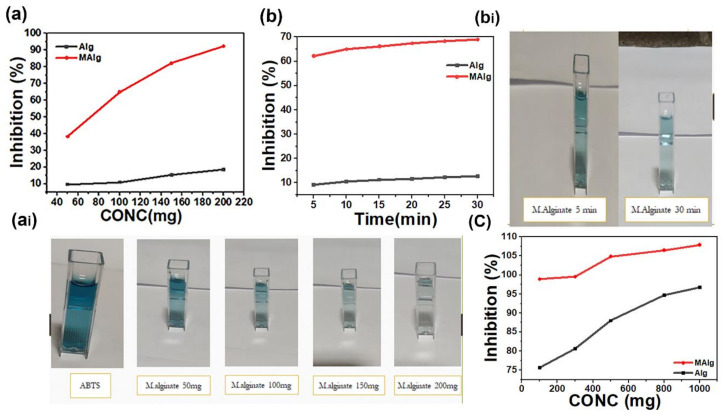
ABTS free radical scavenger activity of Alginic acid and its aromatic derivative. (**a**) with conc. (**ai**) decolorization of ABTS with conc. (**b**) with time. (**bi**) decolonization of ABTS with time. (**c**) scavenger activity of Alginic acid and its aromatic derivative against DPPH radical.

**Figure 9 polymers-13-02575-f009:**
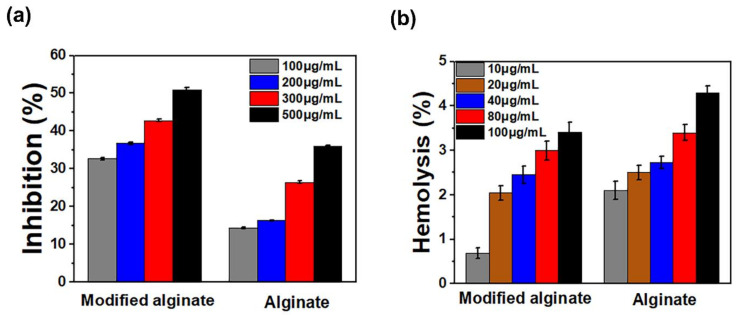
(**a**) Anti-inflammatory properties of alginic acid and its aromatic derivative showing the inhibition of BSA protein denaturation. Each bar indicates mean ± standard deviations of three replications. (**b**) Hemolysis activity of alginate and its aromatic derivative at different polymer concentrations.

**Figure 10 polymers-13-02575-f010:**
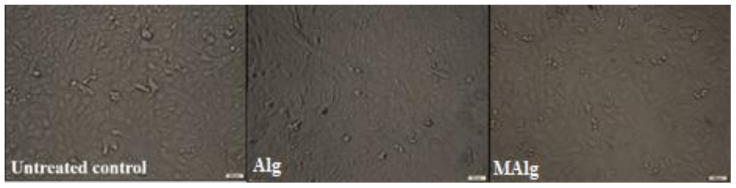
Morphological changes in Wi-38 cells after incubation with the alginic acid and alginate derivative.

**Table 1 polymers-13-02575-t001:** Elemental analysis of alginate and its aromatic derivative.

	C%	H%	C/H%
Alginic acid	36.81	4.984	7.385
Modified Alginate	35.56	5.335	6.665

**Table 2 polymers-13-02575-t002:** Viability % of Wi-38 after 72 h exposure to alginate and aromatic alginate derivative.

Concentration mg/mL	Alginic Acid	Modified Alginate
0.5	66.711 ± 0.026	62.868 ± 0.026
0.25	79.211 ± 1.316	71.921 ± 0.237
0.125	87.026 ± 1.711	83.079 ± 0.184
0.0625	90.895 ± 0.263	89.079 ± 0.237
0.03125	96.263 ± 0.263	95.158 ± 0.421
0.01563	100.632 ± 0.737	99.237 ± 0.395

**Table 3 polymers-13-02575-t003:** Effective concentrations (mg/mL) of alginate and aromatic alginate derivative at 50% and 100% Wi-38 viability.

Concentration mg/mL	Alginic Acid	Modified Alginate
EC50	0.732 ± 0.013	0.630 ± 0.001
EC100	0.065 ± 0.004	0.062 ± 0.001

## Data Availability

There is no data available.

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
