# Peer review of "Enhancement of Antioxidant and Hydrophobic Properties of Alginate via Aromatic Derivatization: Preparation, Characterization, and Evaluation"

_polymers, 2021, doi:10.3390/polym13152575_

Round 1

Reviewer 1 Report

Reviewers' comments:

Manuscript number: polymers-1293632

Title: Enhancement of antioxidant and hydrophobic properties of alginate via aromatic derivatization: preparation, characterization and evaluation.

The manuscript needs a detailed editing. It cannot be recommended for publication in the present form. I hope the following points would be helpful for the authors.

The authors need to consider the following comments

- Line number 17 - derivative at 1431cm-1……...to…..derivative at 1431cm-1.

- Keywords: the authors need to improve with more specific short keywords.

- Provide more detail for introduction part, it is useful for readers.

- Line number 34 - hydrogen peroxide (H2O2), singlet oxygen (1O2),….to…. hydrogen peroxide (H2O2), singlet oxygen (1O2).

- The experimental section should be detailed especially for the Fourier transfer infrared spectroscopy (FT-IR), Elemental Analysis, Thermogravimetric analysis (TGA), Scanning Electron Microscopy (SEM) and Biodegradability. 

- Please provides the references for equations and formula.

- Line number 244 - The new band at 1431 cm-1….to…. The new band at 1431 cm-1.

- Several faults: are added or missing spaces between words (For example – 164, 168, 223, 252, 253, and etc..).

- The authors are obliged to repeat the discussion part of the Differential scanning calorimetry and Morphological analysis.

- In part SEM: how the energy of the accelerator beam used?

- Figure 10 – not clear make clear.

- Invitro blood compatibility – should be improve.

- Some sentences need reconstruction and the level of English should be improved.

- References: make all references in same format for volume number, page number and journal name, because it is difficult to searching and reading.

- Author should provide Graphical Abstract.

So that I recommended this manuscript to major revision and for future process.

Author Response

Thank you for your valuable comments that help us to improve our articles

we take it in our consideration

Reviewer 2 Report

  1. The current study investigates the improvement of several properties of Alginate via Aromatic Derivatization. For this, the authors aim to enhance the Antioxidant and Hydrophobic Properties. The wetting analysis showed that absorption is reduced in the new alginate derivative and showed higher thermostability and improved radical scavenging activity.
  2. The abstract is clear and well written; however, the authors are encouraged to answer the following question: Please consider reviewing the abstract and highlight the novelty, major findings, and conclusions.
  3. Lines 13-14 “as it has a potential activity in biomedical applications” this sentence does not make sense, please rephrase, or try to be clearer of what are you trying to tell us here.
  4. In line 17 what is NHMR? Is it Nuclear magnetic resonance spectroscopy, the authors should mention the full abbreviation first time it appears in the manuscript.
  5. Line 78-87 please combine these small paragraphs into larger ones, please check for this issue everywhere else in the manuscript, the authors are writing very short paragraphs of 2 lines which breaks the continuity and readability of the manuscript.
  6. After line 87 the authors are encouraged to answer the following question: What is the research gap did you find from the previous researchers in your field? Mention it properly. It will improve the strength of the article.
  7. Please combine lines 88-92 in one larger paragraph.
  8. In section 2.1. Materials, the authors should add any available mechanical/chemical/thermal properties in a table format of the basic materials which were used in the current study or any other available information about them.
  9. In section 2.2.1. Preparation of phenolic alginate derivative and following ones, since this is an experimental study, the authors should add some images and figures on the test equipment and setup used in the study and the materials which were fabricated and test equipment used to validate the materials properties..etc. The materials and method section are very dry and just contains text and nothing descriptive which can enrich the visibility of the manuscript.
  10. Line 136 “–800℃” is this minus degree? Please clarify.
  11. For 2.2.8. Biodegradability and 2.2.9.1. ABTS Radical Scavenging Assay and following sections, did the authors follow and standards used for these processes? Mention them it will improve the content of the manuscript.
  12. For some reason, all the degree Celsius symbols are underlines? Please check this issue and amend.
  13. Figure 1 belongs better to the materials and method section; the authors are encouraged to consider moving it to there.
  14. Line 236 “(Scheme 1)” do you mean Figure 1? Please be consistent when referring to Figures in the article.
  15. Move Figure 1 to after line 237 and Figure 2 to after line 245, this way you can better spread the Figures between different paragraphs instead of stacking all the Figures and Tables at the end of the section. It will make the article more organised.
  16. Lines 270-271 “The first stage of weight loss for both Alg and MAlg were lost about…” this sentence needs to be rephrased, perhaps say: In the first stage, the weight reduction (loss) in Alg and MAlg were xx% and xx%, respectively. Or try to rephrase it using your own words.  
  17. Lines 278-279 “indicate to protectivity role of aromatic groups against thermal decomposition” This sentence is not clear as well and needs rephrasing, what are the authors trying to tell us here?
  18. Extensive editing of English language and style required, a lot of the sentence required checking for clarity.
  19. Line 292 “can be explained by Alg ability to hold moisture molecules than MAlg” please support this claim by a reference, how do you know this is true, is it a known fact in the field? If yes then this should be supported by references from the open literature or is it something that was found by past studies, if yes again then it should be referenced properly.
  20. Section 3.4. Morphological analysis is weak, it does not contain any useful information to the readers or any analytical discussion. Please try to expand upon this section and provided meaningful information.
  21. Lines 308-309 please support this finding/claim with references from the open literature, or explain how did you come to find this fact about the distortion of internal order of chains? “presence of heterogeneous molecules between”…”
  22. Please rename Figure 6 to (a) and (b) for clarity and try to separate them with a box or some other tool, when looking at them in the first instance they seemed to be like one Figure.
  23. Line 384-385 “The aromatic alginate derivative was observed to scavenge ABTS•+ in a time and concentration manner” any explanation why this happened? Also was it reported previously in the past literature or did previous studies report something differently from what you found?
  24. Figure 8 the bottom image with different samples is not clear. Please improve the resolution.
  25. Line 408 check the English in this line, it does not read well.
  26. Line 407-408 “, indicating that aromatic alginate derivative shown stronger scavenging activity toward DPPH than alginic acid.” Again, here the authors are stating an interesting finding, but it is not clear why this happened. The authors should attempt to relate this to any explanations found in the open literature or theory on the subject.
  27. Section 3.9. Invitro blood compatibility is a good example of what I need to see in the other sections, some explanation of the observed results or comparison with past work and whether it agrees or disagrees with them, I know it might not be easy to find exactly similar studies to yours but at least find the closest ones related to your work and report what did they find.
  28. For Table 2, the authors should add an extra column at the right side and show the percentage different between the Alginic acid and the modified Aligante. This would better show the difference or consider transforming Table 2 to a bar graph chart figure.
  29. Figure 10 is not clear and does not add any value to the manuscript, it is difficult to make any conclusions from just looking at these three microscopic images.
  30. Conclusion is weak, poorly written and needs to be further expanded. It does not reflect the amount of work done in this paper.
  31. The results in some of the sections are merely described and is limited to comparing the experimental observation. The authors are encouraged to include a discussion in each of these sections and critically discuss the observations from this investigation with existing literature.

Author Response

Thank you for your valuable comments that help us to improve our manuscript.

Round 2

Reviewer 1 Report

The authors have improved the revised manuscript significantly, I recommend acceptance.

Author Response

The authors would like to thank the reviewer for his positive comments and for accepting our revised version

Reviewer 2 Report

  1. Line 347-348 this claim needs to be supported with references and compared against past studies in the open literature.
  2. The English of the manuscript is very poor. I don’t think the authors checked the paper for this issue or if they did then it was not done properly.
  3. Line 387 “Higher free radical scavenging potentials were observed at higher concentrations.” why? This claim/finding must be discussed further and supported with references.

Author Response

  1.  Line 347-348 this claim needs to be supported with references and compared against past studies in the open literature.

Answer

Thank you for your valuable comment. The comparison and reference were added to the text.

  1. The English of the manuscript is very poor. I don’t think the authors checked the paper for this issue, or if they did, then it was not done properly.

Answer

The English of the manuscript was reviewed and improved by an English native checker.

  1. Line 387 “Higher free radical scavenging potentials were observed at higher concentrations.” why? This claim/finding must be discussed further and supported with references.

Answer

Thank you for your valuable comment. The statement meant is normal. As the increase of antioxidant concentration means an increase of functional groups that let to increase the activity.

The sentence was improved and referenced as added to avoid any disturbance to the audient and readers.